

# Investigation of geographic disparities and temporal changes of non-gestational diabetes-related emergency department visits in Florida: a retrospective ecological study

Md Marufuzzaman Khan[1,2,*] and Agricola Odoi[3,*]

[1] Beth Israel Deaconess Medical Center, Boston, MA, United States
[2] Department of Public Health, College of Education, Health, and Human Sciences, University of Tennessee, Knoxville, Tennessee, United States
[3] Biomedical and Diagnostic Sciences, University of Tennessee, Knoxville, Tenneesee, United States
* These authors contributed equally to this work.

Corresponding author
Agricola Odoi, aodoi@utk.edu

## ABSTRACT

**Background:** Rates of diabetes-related Emergency Department (ED) visits in Florida increased by 54% between 2011 and 2016. However, little information is available on geographic disparities of ED visit rates and how these disparities changed over time in Florida and yet this information is important for guiding resource allocation for diabetes control programs. Therefore, the objectives of this study were to (a) investigate geographic disparities and temporal changes in non-gestational diabetes-related ED visit rates in Florida and (b) identify predictors of geographic disparities in non-gestational diabetes-related ED visit rates.

**Methods:** The ED data for the period between 2016 and 2019 were obtained from the Florida Agency for Healthcare Administration. Records of non-gestational diabetes-related ED visits were extracted using the International Classification of Diseases (ICD)-10 codes. Monthly non-gestational diabetes-related ED visit rates were computed and temporal changes were investigated using the Cochran-Armitage trend test. County-level non-gestational diabetes-related ED visit rates per 100,000 person-years were calculated and their geographic distributions were visualized using choropleth maps. Clusters of counties with high non-gestational diabetes-related ED visit rates were identified using Kulldorff's circular and Tango's flexible spatial scan statistics. Predictors of non-gestational diabetes-related ED visit rates were investigated using negative binomial model. The geographic distributions of significant ($p \leq 0.05$) high-rate clusters and predictors of ED visit rates were displayed on maps.

**Results:** There was a significant ($p < 0.001$) increase in non-gestational diabetes-related ED visit rates from 266 visits per 100,000 person-months in January 2016 to 332 visits per 100,000 person-months in December 2019. Clusters of high non-gestational diabetes-related ED visit rates were identified in the northern and south-central parts of Florida. Counties with high percentages of non-Hispanic Black, current smokers, uninsured, and populations with diabetes had significantly

higher non-gestational diabetes-related ED visit rates, while counties with high percentages of married populations had significantly lower ED visit rates.
**Conclusions:** The study findings confirm geographic disparities of non-gestational diabetes-related ED visit rates in Florida with high-rate areas observed in the rural northern and south-central parts of the state. Specific attention is required to address disparities in counties with high diabetes prevalence, high percentages of non-Hispanic Black, and uninsured populations. These findings are useful for guiding public health efforts geared at reducing disparities and improving diabetes outcomes in Florida.

## INTRODUCTION

Diabetes is a major public health problem with a significant economic burden as evidenced by the fact that the average healthcare expenditure of a patient with diabetes is 2.3 times higher than that of a patient without diabetes (*Yang et al., 2018*). Adults with poorly managed diabetes may suffer from short-term complications such as hypoglycemia, hyperglycemia, diabetic ketoacidosis, and long-term complications such as stroke, heart failure, retinopathy, neuropathy, nephropathy, and diabetes-related foot problems such as gangrene, ulcer, Charcot's foot, and amputation (*Mayo Clinic, 2020*; *Khan et al., 2022*). There is evidence that the risk of Emergency Department (ED) visits is higher among persons with diabetes (68 visits/100 persons) than the national average (42.7 visits/100 persons) (*Centers for Disease Control and Prevention, 2021a*, *2021b*). In 2015, about 24% of all ED visits for patients aged 45 or older involved people with diabetes (*Washington, Andrews & Mutter, 2013*; *Hall, Rui & Schwartzman, 2018*). Out of 130 million ED visits in 2018 in the US, approximately 17 million visits were due to diabetes and diabetes-related complications.

Emergency departments play a vital role at the interface between the population and the healthcare system. Investigating patterns in the use of ED visits can help identify patterns of health resource utilization, identify disease trends and emerging threats, as well as assess the magnitude and management of disease problems (*Kellermann et al., 2013*). Previous studies reported that diabetes-related ED visits varied by age, race, ethnicity, income levels, and types of diabetes-related complications (*Washington, Andrews & Mutter, 2013*; *Ginde, Espinola & Camargo, 2008*; *Menchine et al., 2012*; *Benoit et al., 2020*). Not surprisingly, the southern states of the US, including Florida, have considerably higher diabetes-related ED visits than the rest of the country (*Washington, Andrews & Mutter, 2013*). A study identified this region as the diabetes belt, an area where diabetes prevalence was significantly higher than the rest of the US (*Barker et al., 2011*). However, diabetes management programs are not equitably distributed in this area and significant disparities

exist in the use of available diabetes management programs (*Khan et al., 2021*). Lack of access to appropriate and timely diabetes care for some individuals with diabetes results in poor management of the condition leading to otherwise avoidable diabetes-related ED visits resulting in geographic disparities in ED visits (*Ricci-Cabello et al., 2010*; *Barker et al., 2011*; *Shrestha, 2012*; *Walker et al., 2014*, *2015*; *Lord, Roberson & Odoi, 2020*). Identifying these disparities is important in guiding health planning for these patients. Although a study investigated geographic disparities in diabetes-related ED visits in the US at the regional level (*Menchine et al., 2012*), no such studies have been done at lower geographic scales and yet this information is important for guiding resource allocation to address the problem at the local level.

Diabetes is considered an ambulatory care sensitive condition (ACSC), a condition in which appropriate ambulatory/outpatient care can prevent complications and the need for ED visits and hospitalizations (*Agency for Healthcare Research and Quality, 2001*). Therefore, disparities in diabetes-related ED visit may indicate differential outpatient care access, continuity, and quality (*Johnson et al., 2012*). Previous studies identified disparities in ED utilization among traditionally underserved groups such as Black, Hispanic, uninsured, and low-income patients (*Sun, Burstin & Brennan, 2003*; *Hong, Baumann & Boudreaux, 2007*). Since non-Hispanic Black and Hispanic populations represent almost half of the Florida population and there is evidence of a 54% increase in diabetes-related ED visits in Florida between 2011 and 2016, investigating temporal changes and geographic disparities in diabetes-related ED visits is necessary for a better understanding of diabetes burden in Florida (*Florida Diabetes Advisory Council, 2019*; *Florida Department of Health, 2022*). Identifying areas with high diabetes-related ED visit rates in Florida could help identify areas with inadequate access to ambulatory care and poor quality of diabetes management (*Dowd et al., 2014*). This knowledge is important for planning programs targeted at improving access to primary diabetes care, reducing the burden of the condition and its complications, and improving population health. In addition, socioeconomic and demographic predictors of disparities in diabetes-related ED visits, if identified, would help guide resource allocation geared towards reducing disparities in availability of diabetes care as well as diabetes burden in Florida. Therefore, the objectives of this study were to: a) investigate geographic disparities and temporal changes in non-gestational diabetes-related ED visit rates in Florida between 2016 and 2019; b) identify predictors of geographic disparities in non-gestational diabetes-related ED visit rates in Florida.

## MATERIALS AND METHODS

### Study design and area

This retrospective ecological study was conducted between 2022 and 2023. Temporal changes in monthly non-gestational diabetes-related ED visit rates (per 100,000 person-months) were investigated. Annual non-gestational diabetes-related ED visit rates (per 100,000 person-years) were computed at the county level and clusters of counties with high non-gestational diabetes-related ED visit rates were identified and displayed on maps. Socioeconomic and demographic characteristics, health-related behavior, and

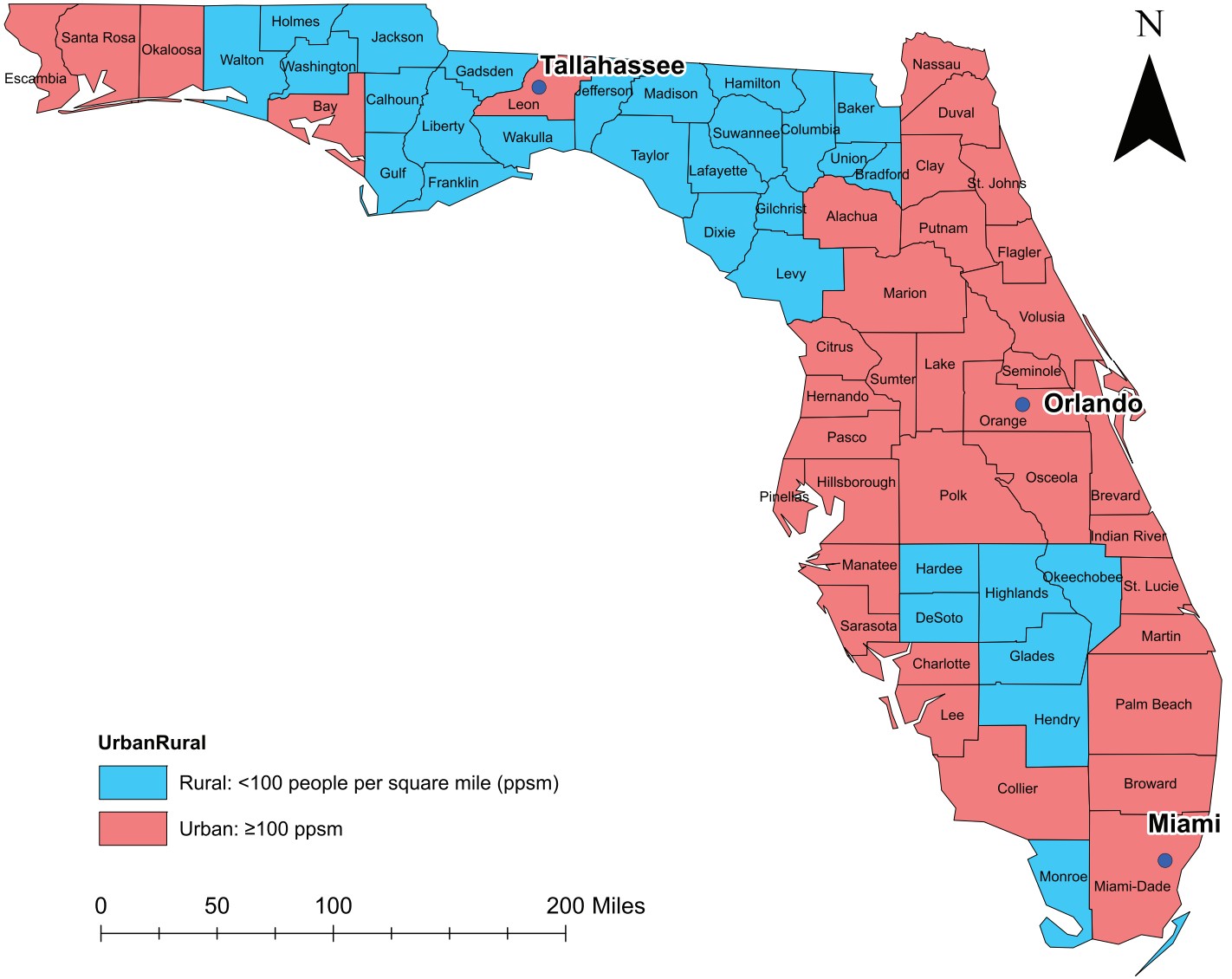

**Figure 1** **Florida map showing geographic distribution of rural and urban counties.** Base maps source: the United States Census Bureau, https://www.census.gov/geographies/mapping-files/time-series/geo/tiger-line-file.2019.html.

environmental factors were investigated as potential predictors of non-gestational diabetes-related ED visit rates.

The study area encompassed the entire state of Florida and covered the time period from January 1, 2016 to December 31, 2019 (Fig. 1). As of 2020, Florida was the most populous state in the southeastern US with approximately 21.6 million people (*Florida Department of Health, 2020a*). Twenty-two percent of Florida's population was 0–19 years old, 31% was 20–44 years old, 26.1% was 45–64 years old, and the rest (20.9%) were 65 years old or older. Approximately half of the population was female. By race, White represented the majority (77.2%) of the population, Black were 17.0%, and all other races comprised the rest (5.8%). By ethnicity, 26.7% of the population was Hispanic-Latino while the rest were non-Hispanic (of any race) (*Florida Department of Health, 2020a*). The

most urban and populous county was Miami-Dade (population: 2.9 million) located in the southern part of the state, while the most rural and least populous county was Lafayette (population: 8,721) located in the northern part of the state (*Florida Department of Health, 2020a*). In total, Florida has 67 counties many of which are considered part of the diabetes belt which is an area located in the southeastern part of the US where diabetes prevalence is higher (11.7%) compared to the rest of the US (8.5%) (*Barker et al., 2011*).

## Data sources

### Emergency Department data

ED data for the time period from January 1, 2016 to December 31, 2019 were obtained from the Agency for Healthcare Administration, Florida. Records of ED visits related to non-gestational diabetes and non-gestational diabetes-related complications, for people of all ages, were extracted from the ED data using the following International Classification of Diseases (ICD)-10 codes listed in primary or secondary diagnoses: E08 (diabetes mellitus due to underlying condition); E09 (drug or chemical induced diabetes mellitus); E10 (type 1 diabetes mellitus); E11 (type 2 diabetes mellitus); and E13 (other specified diabetes mellitus) (*Kostick, 2012*). However, pregnancy related diabetes and neonatal diabetes were excluded.

### Socioeconomic, demographic, health, environmental, and cartographic data

The 2017–2019 Behavioral Risk Factor Surveillance System (BRFSS) data, which contains questionnaire survey data for individuals aged 18 years or older, were obtained from the Florida Department of Health (FDH) (*Florida Department of Health, 2020b*). The following variables were extracted from the BRFSS: respondent's county of residence, race, gender, education, income, marital status, overall health status, Body Mass Index (BMI), level of daily physical activity, daily fruit and vegetable consumption, smoking and drinking habits, usage of tobacco, snuff, or e-cigarettes, healthcare accessibility, presence of comorbidities such as diabetes, heart disease, stroke, arthritis, kidney diseases, hypertension, hypercholesterolemia, and depression.

County-level percentages of population that are unemployed, lack access to healthy food and exercise opportunities, do not have food security, and live in rural areas were obtained from the County Health Rankings and Roadmap (CHRR) website (*Robert Wood Johnson Foundation and The University of Wisconsin Population Health Institute, 2020*). Additionally, county-level number of primary care physicians per 100,000 population and air pollution were extracted from the CHRR website. Data on total number of people per county were obtained from the population dashboard of the FDH (*Florida Department of Health, 2020a*). Rural and urban counties were classified based on the population density in counties. Counties with a population density of <100 persons per square mile were classified as rural, while those with a density of ≥100 were classified as urban. This classification table was obtained from the FDH (*Florida Department of Health, 2023*). County-level age distribution of the population and household vehicle availability data were extracted from the 2015–2019 American Community Survey (ACS) 5-year average estimate (*US Census Bureau, 2020*). A cartographic boundary file for performing

county-level geographic analyses was downloaded from the United States Census Bureau TIGER Geodatabase (*United States Census Bureau, 2021*).

## Data preparation and descriptive analyses

All data preparation and descriptive analyses were performed in SAS 9.4 (*SAS Institute Inc, 2017*). Since the BRFSS data were collected using a complex survey design, all county-level estimates were calculated using SURVEYFREQ procedure of SAS specifying strata variable (_STSTR), cluster variable (_PSU), and sampling weight variable (_CNTYWT). Weighted county-level percentages of categorical variables were computed and presented in a table (Table 1). Annual county-level non-gestational diabetes-related ED visit rates were computed by dividing the number of non-gestational diabetes-related ED visits in a year by the total number of populations of the county and multiplying the result by 100,000.

The Shapiro-Wilk test and Quantile-Quantile (Q-Q) plot were used to assess the normality of continuous county-level variables. Means, standard deviations, median (50th percentile), and lower-upper quartiles were reported for all variables. Temporal trends in monthly diabetes-related ED visit rates were investigated using the Cochran-Armitage trend test.

## Spatial analysis

### Tango's flexible spatial scan statistics

Circular- and irregularly-shaped spatial clusters of high non-gestational diabetes-related ED visit rates were investigated using Tango's flexible spatial scan statistics implemented in FlexScan 3.1.2 (*Tango & Takahashi, 2005*). Poisson probability model with restricted log likelihood (LLR) ratio (specifying $\alpha = 0.2$) and a maximum cluster size of 15 counties was used to ensure that each cluster includes areas with high ED visit rates, based on the geographic distribution of ED visit rates in Florida and prior studies identifying regions with higher disease burdens (*Lord, Roberson & Odoi, 2020*; *Khan et al., 2021*; *Khan, Odoi & Odoi, 2023*). For statistical inference, 999 Monte Carlo replications and a critical *p*-value of 0.05 were used to assess statistical significance. Significant clusters were ranked based on their restricted LLR values. The cluster with the highest LLR value was considered the primary cluster, while the rest of the statistically significant clusters were considered secondary clusters. Only statistically significant non-overlapping clusters that had what we considered an epidemiologically notable higher rate of ED visits compared to the overall rate (*i.e.*, observed/expected ≥1.2) were reported.

### Kulldorff's' circular spatial scan statistics

Kulldorff's circular spatial scan statistics (CSSS), implemented in SaTScan 9.6, was used to identify circular non-overlapping purely spatial high-rate clusters of non-gestational diabetes-related ED visit rates. A discrete Poisson probability model specifying a maximum circular window size of 13.5% of population at risk was used in the analysis. The window size was set based on the geographic distribution of high ED visit rates in Florida, and prior knowledge of areas with high disease rates in Florida (*Lord, Roberson & Odoi, 2020*; *Khan et al., 2021*; *Khan, Odoi & Odoi, 2023*). This window size also ensures that all

**Table 1 Summary statistics of variables considered as potential predictors of county-level non-gestational diabetes-related emergency department visit rates in Florida, 2019.**

| Predictor variable | Mean | SD[1] | Median | IQR[2] | Minimum | Maximum |
|---|---|---|---|---|---|---|
| Percent with age less than 20 years* | 21.68 | 3.40 | 21.70 | 4.20 | 8.30 | 29.50 |
| Percent with age 20 to 44 years | 29.98 | 5.20 | 30.80 | 6.50 | 13.90 | 41.50 |
| Percent with age 45 to 64 years* | 26.69 | 2.09 | 27.00 | 2.20 | 20.80 | 31.70 |
| Percent with age ≥65 years* | 21.64 | 7.73 | 20.10 | 8.40 | 11.60 | 56.70 |
| Percent non-Hispanic White* | 69.75 | 15.02 | 74.07 | 16.05 | 13.07 | 89.53 |
| Percent non-Hispanic Black* | 13.07 | 9.50 | 9.97 | 11.11 | 1.10 | 54.42 |
| Percent Hispanic* | 13.36 | 12.44 | 8.97 | 10.65 | 2.78 | 69.79 |
| Percent non-Hispanic other races* | 3.83 | 1.65 | 3.64 | 2.10 | 0.92 | 8.59 |
| Percent male* | 51.15 | 4.46 | 48.77 | 5.58 | 46.84 | 70.10 |
| Percent female* | 48.86 | 4.46 | 51.23 | 5.58 | 29.91 | 53.16 |
| Percent of having less than high school education* | 15.81 | 6.23 | 14.75 | 7.26 | 5.38 | 38.37 |
| Percent of having high school education | 35.23 | 7.11 | 35.00 | 12.03 | 19.88 | 54.82 |
| Percent of having some college education* | 29.89 | 4.54 | 30.83 | 7.39 | 17.57 | 35.55 |
| Percent of having college education* | 19.07 | 8.06 | 18.44 | 13.51 | 6.06 | 35.62 |
| Percent that income less than 25 k per year* | 33.96 | 7.14 | 34.27 | 12.87 | 20.30 | 53.39 |
| Percent that income 25 to 50 k per year | 27.88 | 4.21 | 28.36 | 5.58 | 20.55 | 40.20 |
| Percent that income more than 50 k per year | 38.15 | 9.41 | 37.17 | 17.12 | 19.39 | 58.94 |
| Percent unemployed* | 3.48 | 0.64 | 3.40 | 0.70 | 2.10 | 5.80 |
| Percent married | 50.75 | 5.27 | 50.38 | 5.96 | 38.49 | 66.98 |
| Percent divorced/widowed/separated | 24.19 | 3.28 | 24.85 | 4.80 | 16.35 | 30.61 |
| Percent never married or unmarried couple* | 25.06 | 5.79 | 23.88 | 7.54 | 10.60 | 45.16 |
| Percent of having overall poor health | 22.56 | 4.86 | 22.59 | 7.48 | 8.59 | 33.13 |
| Percent of having overall good health | 77.44 | 4.86 | 77.41 | 7.48 | 66.88 | 91.41 |
| Percent of being highly active* | 34.53 | 5.61 | 33.78 | 7.08 | 24.35 | 54.60 |
| Percent of being active | 15.27 | 3.45 | 14.86 | 4.46 | 9.09 | 27.53 |
| Percent of being insufficiently active | 15.85 | 3.16 | 15.69 | 3.74 | 9.41 | 26.20 |
| Percent of being inactive | 34.35 | 6.63 | 33.58 | 11.26 | 22.67 | 51.23 |
| Percent of having normal weight | 29.72 | 5.27 | 29.55 | 6.70 | 19.43 | 43.93 |
| Percent of being obese | 32.46 | 6.06 | 32.24 | 8.29 | 18.19 | 48.06 |
| Percent of being overweight | 35.68 | 3.64 | 36.07 | 3.75 | 24.61 | 43.83 |
| Percent of having less than normal weight | 2.14 | 0.91 | 2.12 | 1.29 | 0.31 | 5.39 |
| Percent that eat vegetables ≥once a day | 82.05 | 4.75 | 82.56 | 5.90 | 66.58 | 93.33 |
| Percent that eat fruits ≥once a day | 60.50 | 5.77 | 60.85 | 8.29 | 49.11 | 72.77 |
| Percent that lack access to healthy food* | 9.33 | 5.74 | 9.00 | 6.00 | 0.00 | 31.00 |
| Percent with food insecurity* | 14.00 | 2.22 | 14.00 | 4.00 | 10.00 | 20.00 |
| Percent with access to exercise opportunity* | 68.94 | 24.52 | 77.00 | 36.00 | 10.00 | 100.00 |
| Percent of being current smokers | 19.14 | 5.09 | 18.49 | 6.98 | 11.03 | 32.41 |
| Percent of being current tobacco or snuff user* | 4.99 | 3.27 | 3.58 | 5.21 | 1.24 | 13.52 |
| Percent of being current e-cigarette users* | 5.73 | 1.84 | 5.72 | 2.22 | 2.00 | 13.15 |
| Percent of being heavy drinkers | 7.22 | 2.24 | 6.99 | 2.89 | 1.27 | 12.22 |
| Percent that have no insurance coverage* | 17.39 | 4.32 | 16.76 | 4.87 | 9.45 | 31.45 |

(Continued)

| Predictor variable | Mean | SD[1] | Median | IQR[2] | Minimum | Maximum |
|---|---|---|---|---|---|---|
| Percent that could not see a doctor in the last 12 months | 16.42 | 3.01 | 16.03 | 4.59 | 9.49 | 21.93 |
| Percent that have a personal doctor | 73.65 | 5.12 | 74.39 | 7.14 | 57.61 | 86.03 |
| Number of primary care physician per 100 k population* | 49.93 | 28.13 | 50.77 | 41.82 | 0.00 | 158.26 |
| Percent of houses with no vehicle* | 5.72 | 1.91 | 5.26 | 2.10 | 1.89 | 10.34 |
| Percent of having diabetes | 13.36 | 3.09 | 12.91 | 4.56 | 6.35 | 20.79 |
| Average age of diabetes diagnosis | 48.95 | 2.49 | 49.14 | 3.23 | 42.39 | 53.52 |
| Percent of attending DSME | 53.41 | 10.76 | 53.14 | 16.76 | 29.56 | 76.60 |
| Percent of being depressed | 17.77 | 3.26 | 17.86 | 3.86 | 10.32 | 24.70 |
| Percent that have any disability | 34.35 | 5.39 | 34.65 | 8.30 | 20.99 | 45.86 |
| Percent of having kidney disease* | 3.76 | 1.16 | 3.58 | 1.56 | 1.72 | 7.69 |
| Percent that have regular checkup | 76.11 | 3.88 | 76.09 | 5.09 | 63.19 | 89.08 |
| Percent that take medications for high cholesterol | 61.32 | 5.11 | 61.36 | 7.68 | 47.65 | 70.52 |
| Percent that take medications for hypertension | 78.95 | 4.07 | 78.83 | 5.00 | 67.32 | 89.25 |
| Percent of having myocardial infarction or heart disease | 5.65 | 1.48 | 5.72 | 2.05 | 2.55 | 8.97 |
| Percent of having stroke | 4.52 | 1.29 | 4.51 | 1.94 | 1.23 | 7.01 |
| Percent of having arthritis | 28.97 | 5.31 | 28.74 | 7.10 | 17.80 | 40.21 |
| Percent that have high cholesterol | 32.32 | 3.82 | 31.76 | 4.50 | 23.56 | 43.70 |
| Percent that have hypertension | 38.21 | 5.06 | 37.58 | 7.42 | 25.30 | 46.98 |
| Percent of rural population* | 37.50 | 32.26 | 23.77 | 59.04 | 0.02 | 100.00 |
| Air quality (Average parts per million)* | 7.52 | 0.91 | 7.70 | 1.30 | 5.20 | 9.10 |
| Diabetes-related ED[3] visit rate* (per 100,000 person-years) | 4,342.40 | 1,447.00 | 3,991.11 | 1,498.00 | 1,881.92 | 10,176.10 |

**Notes:**
[1] Standard deviation.
[2] Interquartile range.
[3] Emergency department.
* Non-normally distributed variables.

counties have a chance of being in a cluster regardless of their population size. Like our analyses using Tango's method, the cluster with the highest LLR value was considered the primary cluster, and 999 Monte Carlo replications and a critical $p$-value of 0.05 were used to identify statistically significant clusters, and clusters with rate ratios ≥1.2 were reported.

## Predictors of geographic distributions of non-gestational diabetes-related ED visit rates

For investigation of county-level predictors of non-gestational diabetes-related ED visit rates, only data for 2019 were used to ensure that the identified associations were based on the most current available data. To do this, a global negative binomial model was built in SAS using county-level data obtained from the BRFSS, CHRR, FDH, and ACS (*SAS Institute Inc, 2017*). A causal diagram was constructed based on literature review and biological knowledge to identify potential predictors. The model building process involved first assessing univariable associations between each of the potential predictors and county-level diabetes-related ED visit rates using a relaxed $p$-value of ≤0.15. The log of total number of populations (per 100,000) was used as an offset. Correlations among the

potential predictors were assessed using Spearman's rank correlation coefficients. To avoid multicollinearity, only one of a pair of highly correlated variables ($r \geq |0.7|$) was retained for assessment in the multivariable model. The decision of which variable of the pair to keep was based on its proximal relationship with the outcome as well as its *p*-value (variable with the lowest *p*-value was retained). Variables that were significant potential predictors and not highly correlated were used to build the multivariable negative binomial model. The final main effects model was built using manual backward elimination approach specifying a critical *p*-value of $\leq 0.05$. Potential confounders and intervening variables were identified using the causal diagram and assessed during the model building process. Thus, confounding was assessed by running the model with and without a suspected confounder and assessing the changes in regression coefficients of variables in the model. A variable was kept in the final main effects model as a confounder if its removal from the model resulted in a change of 20% or more of the coefficients of any other variables in the model. Biologically meaningful two-way interaction terms (such as percent non-Hispanic Black*percent of having diabetes; percent non-Hispanic Black*percent without insurance coverage), were assessed and only the statistically significant ones were retained in the final model. Goodness-of-fit of the final model was assessed using deviance $\chi^2$ goodness-of-fit test. Cook's Distance was used to identify highly influential observations.

## Cartographic displays

All cartographic displays were performed using QGIS (*QGIS Development Team, 2021*). The geographic distribution of non-gestational diabetes-related ED visit rates and significant ($p \leq 0.05$) spatial clusters were displayed on maps. Critical intervals for choropleth maps were determined using Jenk's optimization classification scheme.

## Ethics approval

This study was reviewed and approved by the University of Tennessee Institutional Review Board (IRB) (IRB Number: UTK IRB-20-05707-XM). The IRB determined that the study was eligible for exempt review under 45 CFR 46.101 "Category 4: Secondary research for which consent is not required: Secondary research uses of identifiable private information or identifiable biospecimens, if the information, which may include information about biospecimens, is recorded by the investigator in such a manner that the identity of the human subjects cannot readily be ascertained directly or through identifiers linked to the subjects, the investigator does not contact the subjects, and the investigator will not re-identify subjects."

## RESULTS

There was a total of 34,285,646 ED visits reported in Florida from January 1, 2016 to December 31, 2019. This study included only 2,871,326 ED visits, which had a diagnosis of non-gestational diabetes or non-gestational diabetes-related complications.

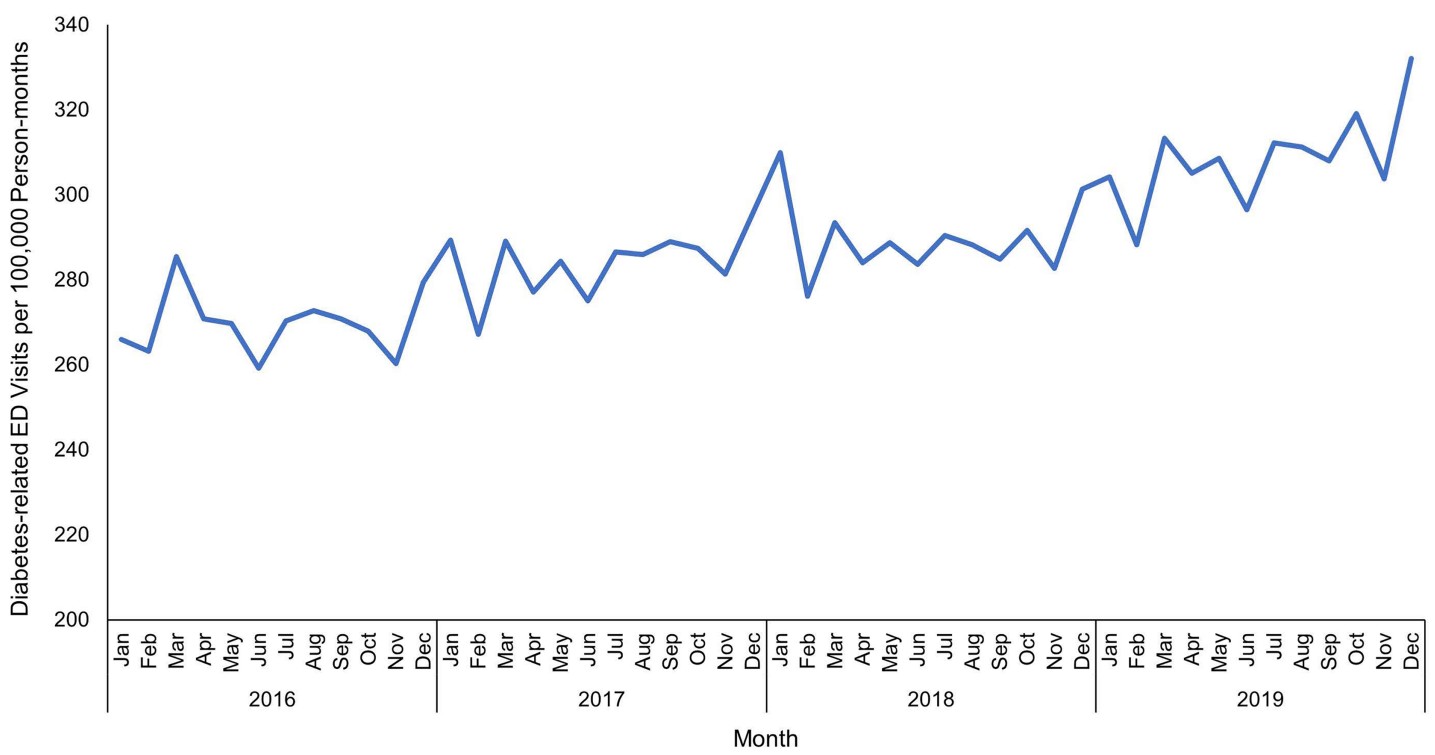

**Figure 2 Temporal patterns of diabetes-related emergency department visit rates in Florida, January 2016-December 2019.**

## Temporal pattern

Overall, non-gestational diabetes-related ED visit rates in Florida increased significantly ($p < 0.001$) from 266 visits per 100,000 person-months in January 2016 to 332 visits per 100,000 person-months in December 2019 (Fig. 2). The highest non-gestational diabetes-related ED visit rate (332 ED visits per 100,000 person-months) was observed in December 2019, while the lowest (259 ED visits per 100,000 person-months) was in June 2016.

## Spatial distribution

Geographic distribution of non-gestational diabetes-related ED visit rates varied across counties in Florida ranging from 1,448 to 10,211 visits per 100,000 person-years (Fig. 3). Overall, more than half of the counties had high non-gestational diabetes-related ED visit rates (>3,385 ED visits per 100,000 person-years) during the study period. Almost all counties in rural northern Florida, including the entire panhandle area up to the westernmost part of the state, had higher non-gestational diabetes-related ED visits than counties in the southern part of the state (Figs. 1 and 3). However, a few counties in the north-central portion had low ED visit rates in 2016 and 2017. Most of the counties in the central part of the state tended to have high non-gestational diabetes-related ED visit rates, while low rates were consistently observed in the southernmost part and urban coastal areas during the study period (Figs. 1 and 3).
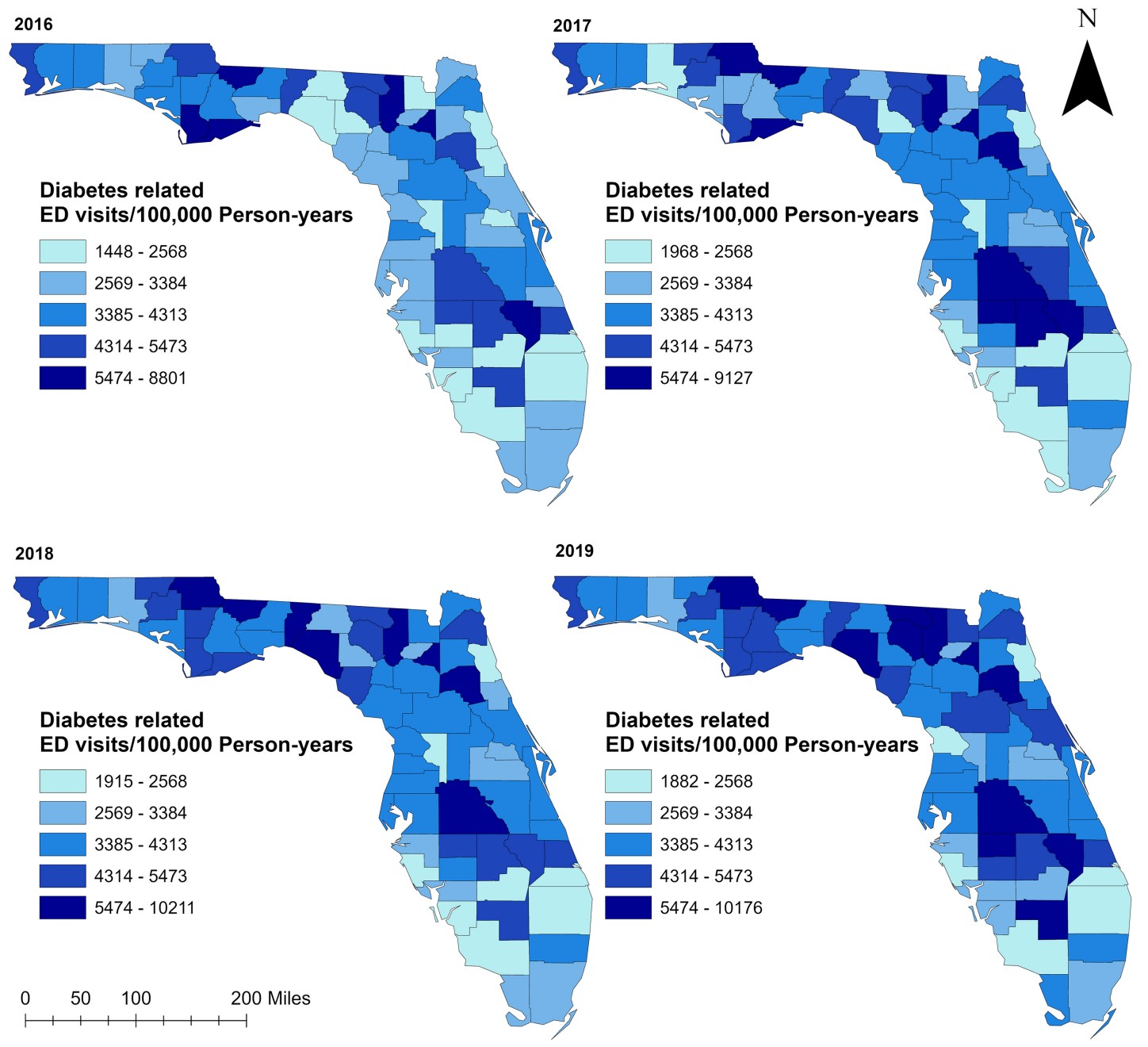

**Figure 3 Geographic distribution of diabetes-related emergency department visit rates in Florida, 2016–2019.** Base maps source: the United States Census Bureau, https://www.census.gov/geographies/mapping-files/time-series/geo/tiger-line-file.2019.html.

## Clusters of high non-gestational diabetes-related ED visit rates

Consistent with high non-gestational diabetes-related ED visit rates observed in the northern and central parts of Florida, significant ($p < 0.05$) high non-gestational diabetes-related ED visit rate clusters were identified in these areas (Figs. 3, 4, and 5). Overall, clusters were similar in size and location across years (Tables 2 and 3, Figs. 4 and 5). Using Tango's FSSS, the primary high-rate clusters were consistently identified in the

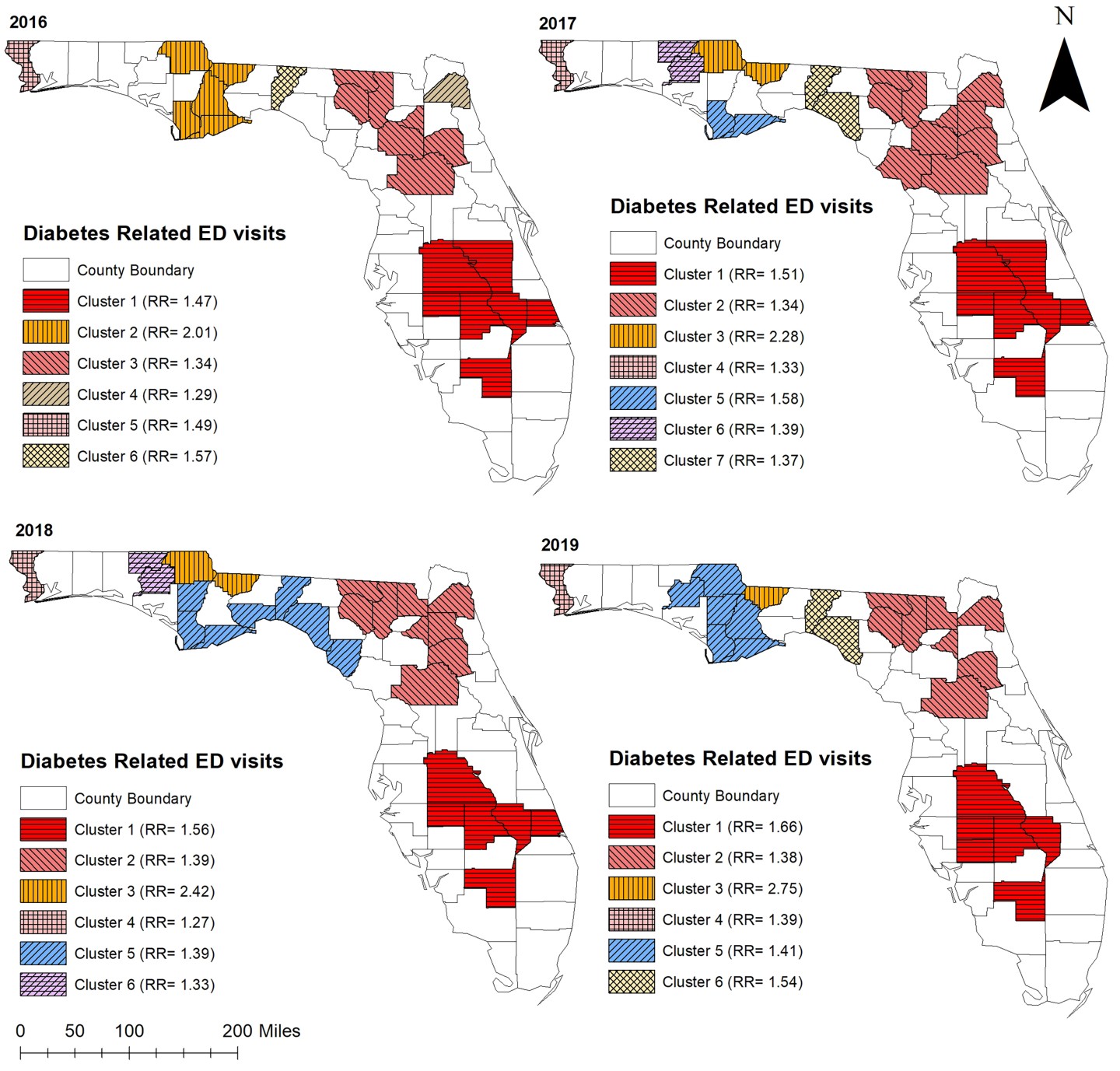

**Figure 4 Statistically significant non-overlapping spatial clusters of high diabetes-related emergency department visit rates identified in Florida using Tango's flexible spatial scan statistics, 2016–2019.** Base maps source: the United States Census Bureau, https://www.census.gov/geographies/mapping-files/time-series/geo/tiger-line-file.2019.html.

south-central portion of the state and mainly included rural counties (Hardee, Highlands, Okeechobee, and Hendry) (Figs. 1 and 4). Similarly, several small high-rate clusters located across the panhandle area included only rural counties. However, a high-rate cluster was identified in the rural-urban interface of north-central Florida and included both rural and

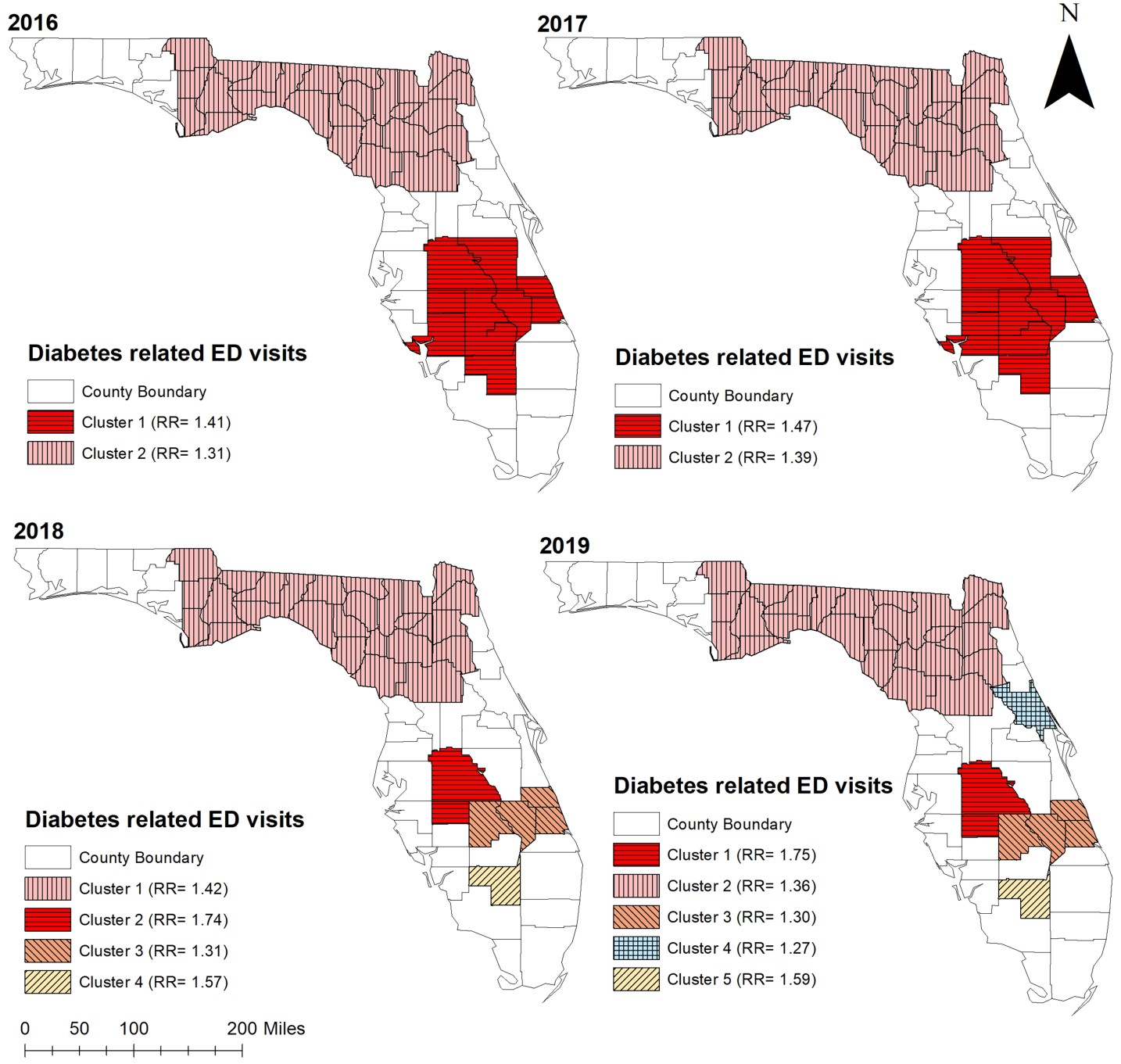

**Figure 5 Statistically significant non-overlapping spatial clusters of high diabetes-related emergency department visit rates identified in Florida using Kulldorff's circular spatial scan statistics, 2016–2019.** Base maps source: the United States Census Bureau, https://www.census.gov/geographies/mapping-files/time-series/geo/tiger-line-file.2019.html.     

urban counties. Although high-rate clusters were not identified in the southern and coastal urban areas, a single county high-rate cluster (Escambia) was consistently identified in the westernmost urban part of the state (Figs. 1 and 4). Similar to the findings of Tango's FSSS, Kulldorff's CSSS identified significant high non-gestational diabetes-related ED visit rate

**Table 2 Statistically significant non-overlapping spatial clusters of high diabetes-related emergency department visit rates identified in Florida using Tango's flexible spatial scan statistics, 2016–2019.**

| Year | Cluster* | Population | Observed ED[1] visits | Expected ED visits | No. of counties | RR[2] | p-value[3] |
|------|----------|-----------|----------------------|--------------------|-----------------|-------|-----------|
| 2016 | Cluster 1 | 1,578,659 | 70,233 | 47,893 | 7 | 1.47 | 0.001 |
|      | Cluster 2 | 135,942 | 8,871 | 4,409 | 5 | 2.01 | 0.001 |
|      | Cluster 3 | 859,052 | 36,193 | 26,947 | 7 | 1.34 | 0.001 |
|      | Cluster 4 | 971,842 | 38,837 | 30,031 | 1 | 1.29 | 0.001 |
|      | Cluster 5 | 322,901 | 15,028 | 10,054 | 1 | 1.49 | 0.001 |
|      | Cluster 6 | 14,842 | 736 | 469 | 1 | 1.57 | 0.001 |
| 2017 | Cluster 1 | 1,578,659 | 77,918 | 51,596 | 7 | 1.51 | 0.001 |
|      | Cluster 2 | 2,089,357 | 93,015 | 69,367 | 10 | 1.34 | 0.001 |
|      | Cluster 3 | 98,251 | 7,676 | 3,374 | 2 | 2.28 | 0.001 |
|      | Cluster 4 | 322,901 | 14,129 | 10,661 | 1 | 1.33 | 0.001 |
|      | Cluster 5 | 28,524 | 1,557 | 987 | 2 | 1.58 | 0.001 |
|      | Cluster 6 | 45,565 | 2,128 | 1,536 | 2 | 1.39 | 0.001 |
|      | Cluster 7 | 37,494 | 1,721 | 1,253 | 2 | 1.37 | 0.001 |
| 2018 | Cluster 1 | 1,209,981 | 65,020 | 41,651 | 6 | 1.56 | 0.001 |
|      | Cluster 2 | 1,809,443 | 85,977 | 61,927 | 9 | 1.39 | 0.001 |
|      | Cluster 3 | 98,251 | 8,324 | 3,436 | 2 | 2.42 | 0.001 |
|      | Cluster 4 | 322,901 | 14,011 | 11,020 | 1 | 1.27 | 0.001 |
|      | Cluster 5 | 129,934 | 6,300 | 4,519 | 7 | 1.39 | 0.001 |
|      | Cluster 6 | 45,565 | 2,106 | 1,587 | 2 | 1.33 | 0.001 |
| 2019 | Cluster 1 | 936,626 | 57,579 | 34,683 | 6 | 1.66 | 0.001 |
|      | Cluster 2 | 1,592,334 | 81,540 | 58,963 | 8 | 1.38 | 0.001 |
|      | Cluster 3 | 47,926 | 4,877 | 1,775 | 1 | 2.75 | 0.001 |
|      | Cluster 4 | 322,901 | 16,641 | 11,957 | 1 | 1.39 | 0.001 |
|      | Cluster 5 | 128,345 | 6,717 | 4,753 | 6 | 1.41 | 0.001 |
|      | Cluster 6 | 37,494 | 2,137 | 1,388 | 2 | 1.54 | 0.001 |

Notes:
[1] Emergency department.
[2] Rate ratio = Observed ED visits/Expected ED visits in the potential cluster.
[3] Statistical significance was assessed using a critical $p = 0.05$.
* Cluster 1: Primary Cluster; Cluster ≥2: Secondary Cluster.
Spatial scan parameters: Maximum cluster size = 15 counties; Test = one-tailed (high rates); RR for reporting clusters ≥ 1.2.
Geographic distribution of the clusters is shown in Fig. 4.

clusters in the panhandle area, north-central, and central portions of the state and included mostly rural counties. However, unlike Tango's FSSS, larger but fewer clusters were identified by Kulldorff's CSSS (Tables 2 and 3, Figs. 4 and 5).

## Predictors of non-gestational diabetes-related ED visit rates

Tables 4 and 5 show the results of the univariable and final multivariable negative binomial models used to investigate associations between county-level sociodemographic variables and county-level non-gestational diabetes-related ED visit rates, respectively. Several of the variables assessed had univariable statistically significant associations with the outcome based on a relaxed critical $p$-value of 0.15. Almost half of these variables were dropped during the variable selection process due to the presence of high collinearity and the fact

**Table 3 Statistically significant non-overlapping spatial clusters of high diabetes-related emergency department visit rates identified in Florida using Kulldorff's circular spatial scan statistics, 2016–2019.**

| Year | Cluster* | Population | Observed ED[1] visits | Expected ED visits | No. of counties | RR[2] | p-value[3] |
|---|---|---|---|---|---|---|---|
| 2016 | Cluster 1 | 1,846,525 | 81,070 | 59,761 | 11 | 1.41 | <0.001 |
| | Cluster 2 | 2,698,053 | 109,703 | 87,319 | 27 | 1.31 | <0.001 |
| 2017 | Cluster 1 | 1,886,441 | 90,568 | 64,295 | 11 | 1.47 | <0.001 |
| | Cluster 2 | 2,731,991 | 122,774 | 93,114 | 27 | 1.39 | <0.001 |
| 2018 | Cluster 1 | 2,759,367 | 129,329 | 95,906 | 27 | 1.42 | <0.001 |
| | Cluster 2 | 709,127 | 41,802 | 24,647 | 2 | 1.74 | <0.001 |
| | Cluster 3 | 601,631 | 27,223 | 20,911 | 4 | 1.31 | <0.001 |
| | Cluster 4 | 39,682 | 2,160 | 1,379 | 1 | 1.57 | <0.001 |
| 2019 | Cluster 1 | 716,081 | 45,210 | 26,516 | 2 | 1.75 | <0.001 |
| | Cluster 2 | 2,798,463 | 134,345 | 103,625 | 27 | 1.36 | <0.001 |
| | Cluster 3 | 609,119 | 29,056 | 22,555 | 4 | 1.30 | <0.001 |
| | Cluster 4 | 539,563 | 25,178 | 19,980 | 1 | 1.27 | <0.001 |
| | Cluster 5 | 40,089 | 2,353 | 1,484 | 1 | 1.59 | <0.001 |

**Notes:**
[1] Emergency department.
[2] Rate ratio = ED visit rate inside the potential cluster/ED visit rate outside the potential cluster.
[3] Statistical significance was assessed using a critical $p = 0.05$.
* Cluster 1: Primary Cluster; Cluster ≥2: Secondary Cluster.
Spatial scan parameters: Maximum window size = 13.5% of total population; Test = one-tailed (high rates); RR for reporting clusters ≥ 1.2.
Geographic distribution of the clusters is presented in Fig. 5.

**Table 4 Results of univariable negative binomial regression models examining the associations between county characteristics and the rate of diabetes-related emergency department visits in Florida, 2019.**

| Predictor variable | IRR[1] (95% CI[2]) | p-value[3] |
|---|---|---|
| Percent with age less than 20 years | 1.034 [1.014–1.056] | 0.001 |
| Percent with age 20 to 44 years | 1.020 [1.004–1.035] | 0.012 |
| Percent with age 45 to 64 years | 0.984 [0.947–1.022] | 0.397 |
| Percent with age ≥65 years | 0.986 [0.976–0.995] | 0.003 |
| Percent non-Hispanic White | 0.994 [0.989–0.999] | 0.023 |
| Percent non-Hispanic Black | 1.015 [1.007–1.022] | <0.001 |
| Percent Hispanic | 0.999 [0.993–1.005] | 0.743 |
| Percent non-Hispanic other races | 0.967 [0.924–1.013] | 0.155 |
| Percent male | 1.017 [0.998–1.036] | 0.073 |
| Percent female | 0.983 [0.966–1.002] | 0.073 |
| Percent of having less than high school education | 1.035 [1.025–1.045] | <0.001 |
| Percent of having high school education | 1.023 [1.012–1.033] | <0.001 |
| Percent of having some college education | 0.964 [0.950–0.978] | <0.001 |
| Percent of having college education | 0.975 [0.968–0.982] | <0.001 |
| Percent that income less than 25 k per year | 1.030 [1.021–1.039] | <0.001 |
| Percent that income 25 to 50 k per year | 1.026 [1.008–1.045] | 0.005 |

(Continued)

| Predictor variable | IRR[1] (95% CI[2]) | p-value[3] |
|---|---|---|
| Percent that income more than 50 k per year | 0.978 [0.972–0.984] | <0.001 |
| Percent unemployed | 1.227 [1.092–1.379] | <0.001 |
| Percent married | 0.975 [0.962–0.988] | <0.001 |
| Percent divorced/widowed/separated | 1.026 [1.003–1.049] | 0.028 |
| Percent never married or unmarried couple | 1.014 [1.000–1.028] | 0.050 |
| Percent of having overall poor health | 1.041 [1.028–1.054] | <0.001 |
| Percent of having overall good health | 0.961 [0.949–0.972] | <0.001 |
| Percent of being highly active | 0.970 [0.959–0.982] | <0.001 |
| Percent of being active | 0.969 [0.949–0.990] | 0.003 |
| Percent of being insufficiently active | 0.990 [0.965–1.015] | 0.417 |
| Percent of being inactive | 1.032 [1.022–1.041] | <0.001 |
| Percent of having normal weight | 0.969 [0.956–0.982] | <0.001 |
| Percent of being obese | 1.038 [1.028–1.048] | <0.001 |
| Percent of being overweight | 0.962 [0.945–0.980] | <0.001 |
| Percent of having less than normal weight | 0.998 [0.919–1.084] | 0.960 |
| Percent that eat vegetables ≥once a day | 0.980 [0.964–0.995] | 0.011 |
| Percent that eat fruits ≥once a day | 0.980 [0.967–0.992] | 0.002 |
| Percent that lack access of healthy food | 1.009 [0.996–1.023] | 0.167 |
| Percent with food insecurity | 1.010 [1.068–1.132] | <0.001 |
| Percent that have access of exercise opportunity | 0.995 [0.992–0.999] | 0.005 |
| Percent of being current smokers | 1.025 [1.010–1.041] | 0.001 |
| Percent of being current tobacco or snuff user | 1.029 [1.006–1.054] | 0.016 |
| Percent of being current ecig users | 0.990 [0.947–1.034] | 0.637 |
| Percent of being heavy drinkers | 0.944 [0.915–0.975] | <0.001 |
| Percent without insurance coverage | 1.042 [1.026–1.059] | <0.001 |
| Percent that could not see a doctor in the last 12 months | 1.048 [1.024–1.073] | <0.001 |
| Percent that have a personal doctor | 0.992 [0.977–1.007] | 0.289 |
| Number of primary care physician per 100 k population | 0.995 [0.992–0.997] | <0.001 |
| Percent of houses with no vehicle | 1.060 [1.022–1.100] | 0.002 |
| Percent of having diabetes | 1.054 [1.031–1.078] | <0.001 |
| Average age of diabetes diagnosis | 0.961 [0.932–0.990] | 0.008 |
| Percent of attending DSME | 0.989 [0.983–0.996] | 0.001 |
| Percent of being depressed | 1.009 [0.986–1.032] | 0.455 |
| Percent that have any disability | 1.028 [1.015–1.042] | <0.001 |
| Percent of having kidney disease | 1.078 [1.011–1.149] | 0.023 |
| Percent that have regular checkup | 0.984 [0.964–1.005] | 0.128 |
| Percent that take medications for high cholesterol | 1.010 [0.994–1.025] | 0.222 |
| Percent that take medications for hypertension | 0.985 [0.965–1.005] | 0.130 |
| Percent of having myocardial infarction or heart disease | 1.042 [0.989–1.098] | 0.119 |
| Percent of having stroke | 1.076 [1.016–1.140] | 0.012 |
| Percent of having arthritis | 1.000 [0.985–1.015] | 1.000 |

| Predictor variable | IRR[1] (95% CI[2]) | p-value[3] |
|---|---|---|
| Percent that have high cholesterol | 0.989 [0.969–1.010] | 0.299 |
| Percent of having hypertension | 1.019 [1.005–1.033] | 0.009 |
| Percent of rural population | 1.004 [1.002–1.007] | <0.001 |
| Air quality (Average parts per million) | 1.089 [1.003–1.183] | 0.042 |

Notes:
[1] Incidence rate ratio.
[2] Confidence interval.
[3] Statistical significance was assessed using a critical $p \leq 0.15$.

**Table 5 Results of a multivariable negative binomial regression model examining the associations between county characteristics and the rate of diabetes-related emergency department visits in Florida, 2019.**

| Predictor variable | IRR[1] (95% CI[2]) | p-value[3] |
|---|---|---|
| Percent non-Hispanic Black | 1.010 [1.004–1.016] | <0.001 |
| Percent of having diabetes | 1.036 [1.018–1.054] | <0.001 |
| Percent without insurance coverage | 1.019 [1.007–1.032] | 0.003 |
| Percent current smokers | 1.019 [1.009–1.030] | <0.001 |
| Percent married | 0.986 [0.975–0.997] | 0.010 |

Notes:
[1] Incidence rate ratio.
[2] Confidence interval.
[3] Statistical significance was assessed using a critical $p \leq 0.05$.

that none of the dropped variables were important confounders. Additionally, no statistically significant two-way interactions were identified. Thus, only variables that had statistically significant ($p < 0.05$) associations with non-gestational diabetes related ED visit rates were kept in the final model.

Based on the final multivariable negative binomial model, the following independent variables had significant high county-level non-gestational diabetes-related ED visit rates: percentages of population who were non-Hispanic Black, current smokers, had diabetes, and had no insurance coverage (Table 5). However, percentage of population that were married had statistically significant low diabetes-related ED visit rate (Table 5). The $p$-value of the deviance $\chi^2$ goodness-of-fit test was not significant ($p = 0.259$) indicating a good model fit. Based on the Cook's Distance, Glades and Union counties were identified as highly influential observations. Glades county had the highest diabetes prevalence, while Union County had the highest percentage of smokers.

Geographic distributions of the significant predictors of non-gestational diabetes-related ED visit rates are shown in Fig. 6. Counties in the central and eastern panhandle rural area tended to have high percentages of population that were non-Hispanic Black, current smokers, had diabetes, and had no insurance coverage. These counties overlapped with many counties with high non-gestational diabetes-related ED visit rates (Figs. 1, 3, and 6). On the other hand, counties in the southernmost urban area of the state had relatively low non-gestational diabetes-related ED visit rates but tended to have high

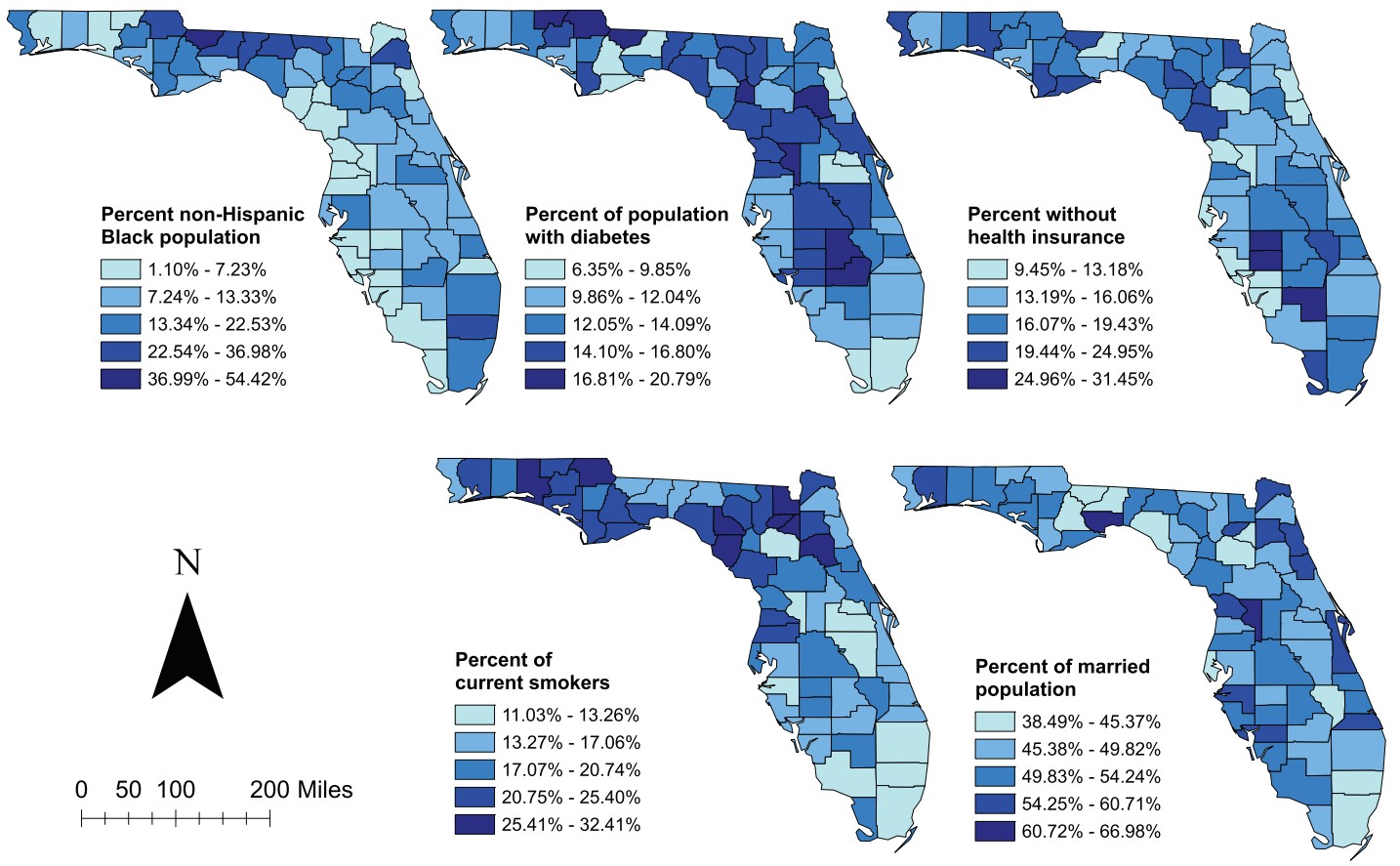

**Figure 6  Distribution of statistically significant predictors of diabetes-related emergency department visit rates in Florida, 2019.** Base maps source: the United States Census Bureau, https://www.census.gov/geographies/mapping-files/time-series/geo/tiger-line-file.2019.html.

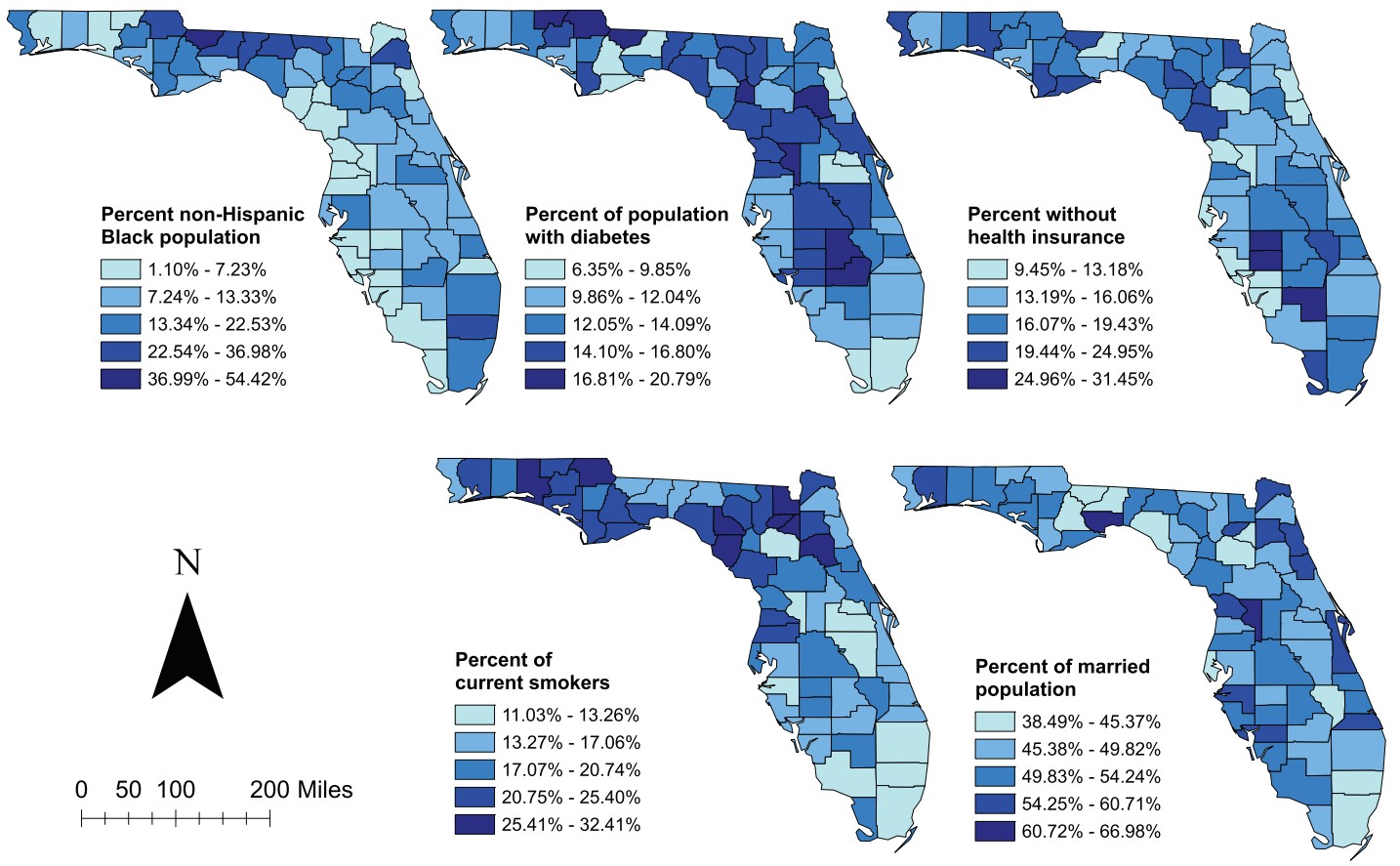

percentages of non-Hispanic Black and low percentages of married, current smokers, and those with diabetes. Although counties with high percentages of population with diabetes were concentrated in the mid to south-central parts of the state, these counties had high percentages of uninsured people (Figs. 1, 3, and 6).

## DISCUSSION

This study investigated geographic disparities and temporal changes of non-gestational diabetes-related ED visit rates in Florida from 2016 to 2019. Although diabetes prevalence and healthcare access for individuals with diabetes vary across counties in Florida (*Ricci-Cabello et al., 2010*; *Barker et al., 2011*; *Lord, Roberson & Odoi, 2020*), little is known about the geographic disparities of non-gestational diabetes-related ED visits and yet this information is critical for improving the health and quality of life of populations with diabetes. The findings of the current study help to fill this gap and are important for guiding healthcare planning targeted at reducing disparities in non-gestational diabetes-related ED visit rates in Florida. In addition, this study investigated sociodemographic, environmental, and lifestyle-related predictors of non-gestational diabetes-related ED visit rates. The results are useful for guiding evidence-based resource allocation aimed at

guiding the implementation of control programs and reducing the burden of diabetes in Florida.

The observed high non-gestational diabetes-related ED visit rates in the northern and southcentral parts of Florida may be related to access to diabetes care in these rural communities. Since diabetes is an ambulatory-care sensitive condition, getting regular primary care could substantially reduce ED visit risks and improve quality of life (*Agency for Healthcare Research and Quality, 2001*). However, access to primary care for diabetes could be limited due to the lack of health insurance coverage, which could result in higher diabetes-related ED visit rates. This is evidenced by the findings of this study because counties with high percentages of populations with no health insurance coverage had high diabetes-related ED visit rates. Additionally, a study by *Khan et al. (2021)* reported that northern Florida had lower diabetes self-management education (DSME) program participation rates than urban areas due to fewer DSME centers and limited accessibility to DSME program. The DSME program was developed to educate diabetes patients on disease management and reduce diabetes-related complications and ED visit rates (*Powers et al., 2015*). However, lack of health insurance coverage among populations in northern Florida might have prevented access to DSME programs, lowered DSME participation rates, and, therefore, resulted in higher ED visit rates.

Another reason of the high non-gestational diabetes-related ED visit rates in northern Florida could be the high percentages of non-Hispanic Black populations in those areas. The findings of this study showed that counties with high percentages of non-Hispanic Black populations tended to have significantly high ED visit rates, which is consistent with the findings of previous studies (*Taylor et al., 2017*; *Uppal et al., 2022*). A study by *Uppal et al. (2022)* reported that diabetes-specific ED use among non-Hispanic Black patients was approximately 3 times higher than among non-Hispanic White patients. This difference was the result of non-Hispanic Black patients having higher risks of diabetes-related complications such as albuminuria, retinopathy, lower extremity amputation, end-stage renal disease (ESRD), and worse glycemic control than their non-Hispanic White counterparts (*Osborn, De Groot & Wagner, 2013*; *Canedo et al., 2018*). Moreover, minority groups, such as Black and Hispanic populations, are less likely to receive recommended diabetes preventive care (*Taylor et al., 2017*) and have low participation in DSME programs (*Khan et al., 2021*).

The significant low county-level non-gestational diabetes-related ED visit rates among the percentage of married people identified in this study suggests that social support from marriage relationships may be beneficial for diabetes patients. Previous studies reported lower diabetes morbidity and mortality among married persons compared to their unmarried counterparts (*Kposowa, Ezzat & Breault, 2021*). This relationship is explained by the fact that married people are more likely to get better social and mental support (*Umberson, 1992*), lead healthy lifestyles (*Eng et al., 2005*), and have better medication adherence and diabetes management (*Gelaw et al., 2014*; *Ahmed, Abugalambo & Almethen, 2017*).

The finding that counties with high percentages of current smokers had higher non-gestational diabetes-related ED visit rates is consistent with reports from previous

studies (*U.S. Department of Health and Human Services, 2010*, *2014*). According to a report published by the Centers for Disease Control and Prevention (CDC), people who smoke and have diabetes are more likely to develop serious health problems from diabetes such as heart disease, kidney disease, retinopathy, peripheral neuropathy, and lower leg amputations (*U.S. Department of Health and Human Services, 2010*; *Centers for Disease Control and Prevention, 2022*). Moreover, these people tend to have trouble with insulin dosing and managing diabetes (*U.S. Department of Health and Human Services, 2010*, *2014*).

## STRENGTHS AND LIMITATIONS

This is the first study investigating geographic disparities of non-gestational diabetes-related ED visits in Florida using rigorous statistical approaches. Identifying areas with high diabetes-related ED visit rates is crucial for guiding resource allocation and improving access to primary diabetes care. This study also investigated sociodemographic predictors of non-gestational diabetes-related ED visits in Florida, the findings of which are important for guiding programs aimed at reducing disparities in the availability of diabetes care and improving the health of populations with diabetes in Florida. Although this study was conducted in Florida, the findings of this study showed that spatial statistics are useful for identifying geographic disparities of ED visit rates across states in the US. However, this study is not without limitation. Coding errors might have occurred in the use of ICD-10 codes to report diabetes and diabetes-related conditions in ED data due to the complex nature of the ICD-10 coding system that might result in coding errors potentially leading to misclassification of the outcome and hence misclassification bias (*American Academy of Family Physicians, 2013*; *Burles et al., 2017*). Similarly, errors in identifying geographic location might have resulted in location misclassification bias. However, this is expected to be minimal, if at all present, and is expected to have little to no impact on study findings since only aggregated county-level data were used. Additionally, since this was a retrospective study that used previously collected administrative data, the investigation was limited to only variables that were available in the datasets used for the study. It is important to note that since this study was conducted at the county-level, it is potentially subject to ecological bias so our inferences to the individual-level should be considered with caution. Finally, BRFSS data used in this study may be prone to reporting bias. However, previous research has shown that findings generated from BRFSS data are representative of the population (*Stein et al., 1995*; *Bowlin et al., 1996*). These limitations notwithstanding, the findings of this study provide useful information for guiding health planners in allocating resources and reducing diabetes burden in Florida.

## CONCLUSIONS

This study identified geographic disparities of non-gestational diabetes-related ED visit rates in Florida with high-rate areas being observed in the rural northern and southcentral parts of the state. Lack of healthcare access, high diabetes prevalence, low levels of insurance coverage, and certain demographic factors were identified as significant predictors of high non-gestational diabetes-related ED visit rates. These findings are useful

for guiding public health efforts geared at reducing disparities and improving diabetes outcomes in Florida.

## ABBREVIATIONS

| | |
|---|---|
| **ED** | Emergency Department |
| **ICD** | International Classification of Diseases |
| **US** | United States |
| **ACSC** | Ambulatory Care Sensitive Condition |
| **FDH** | Florida Department of Health |
| **BRFSS** | Behavioral Risk Factor Surveillance System |
| **BMI** | Body Mass Index |
| **CHRR** | County Health Rankings and Roadmap |
| **ACS** | American Community Survey |
| **FSSS** | Flexible Spatial Scan Statistics |
| **CSSS** | Circular Spatial Scan Statistics |
| **DSME** | Diabetes Self-Management Education |
| **CDC** | Centers for Disease Control and Prevention. |

## ACKNOWLEDGEMENTS

The authors are grateful to the Florida Department of Health for providing the data for this study.

### Funding
The authors received no funding for this work.

### Competing Interests
Agricola Odoi is an Academic Editor for PeerJ.

### Author Contributions

- Md Marufuzzaman Khan conceived and designed the experiments, performed the experiments, analyzed the data, prepared figures and/or tables, authored or reviewed drafts of the article, and approved the final draft.
- Agricola Odoi conceived and designed the experiments, performed the experiments, analyzed the data, prepared figures and/or tables, authored or reviewed drafts of the article, and approved the final draft.

### Human Ethics
The following information was supplied relating to ethical approvals (*i.e.*, approving body and any reference numbers):

University of Tennessee Institutional Review Board (UTK IRB-20-05707-XM).

## Data Availability

The raw data are available in the Supplemental File.

## Supplemental Information

Supplemental information for this article can be found online at http://dx.doi.org/10.7717/peerj.18897#supplemental-information.

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
