# Peer review of "Investigation of geographic disparities and temporal changes of non-gestational diabetes-related emergency department visits in Florida: a retrospective ecological study"

_PeerJ, doi:10.7717/peerj.18897_

## Round 0.1 · original submission · Minor Revisions

Dear Dr. Odoi,

Please resubmit a revised version of your manuscript with a complete response to each of the reviewers' comments.

Yours,

Yoshi

Prof. Yoshinori Marunaka, M.D., Ph.D.

Reviewer 1 ·

Basic reporting

Yes (see full review in additional comments).

Experimental design

Yes (see full review in additional comments).

Validity of the findings

Yes (see full review in additional comments).

Additional comments

General comments:

Overall, this is a very interesting and well written manuscript with important results for public health researchers and policy makers focused on controlling complications of diabetes mellitus. However, there are a few areas of the current manuscript that require correction, elaboration, and/or justification. The following should be considered by the authors in revising their manuscript:
1. The authors use the term “risk” throughout the manuscript but appear to be identifying spatial clusters of high rates and variables associated with rates of diabetes-related emergency department (ED) visits. This should be changed in the revised draft including in tables and figures.
2. The authors should be specific throughout the manuscript, including the title and abstract, that they are examining factors that influence rates of ED visits due to non-gestational diabetes mellitus. I had assumed this was the case, but this was only confirmed once I read the methods section of the manuscript concerning the form of diabetes being studied.
3. It is unclear why Poisson or negative binomial models were not fitted to analyze factors associated with rates of diabetes-related emergency department (ED) visits. Based on the results concerning their diagnostics, I suspect the same results were obtained with their linear regression model (i.e., the residuals met the assumptions of normality, homoskedasticity, and independence), but it seems a bit odd to use Poisson models for their scans and then a normal distribution for their regression models. Perhaps the authors could elaborate on this decision in the methods and/or discussion. It was also a bit unclear why year was not included in the modeling section of the manuscript.
4. In the limitations section, the authors should briefly address ecological bias, potential selection bias, and consider misclassification bias of location in addition to ICD-10 classification.
5. The authors note that they set “restrictions” on the size of potential clusters (i.e., below standard defaults). In effect, I believe this decision has artificially created a number of small adjacent spatial clusters when in fact there are probably just two major clusters (i.e., in the panhandle and centre of the state). While setting restrictions on sizes of clusters can have epidemiological/biological justification, some of the explanations might need more elaboration. For instance, the maximum size for Tango’s flexible spatial scan statistic was set to 15 counties to avoid including counties with non-elevated risks. It is a bit unclear how this setting would accomplish that goal. Similarly, for Kulldorff’s spatial scan statistic, the maximum window was set at 13.5% of the population based on the population of the largest county in Florida to ensure that all counties have a chance of being a cluster. However, scan statistics can identify smaller clusters if they are statistically significant and using this setting prevented Miami-Dade County from being part of a larger cluster. It would be great if the authors could elaborate more on these decisions in the discussion and/or comment on the impact of using these restrictions compared to the default options.
Specific comments:

Title:
i. As noted above, clarify that this manuscript focuses on non-gestational diabetes mellitus.

Abstract:
i. Please see general comments concerning “diabetes mellitus” and “rates”.

Key words:
i. You appear to have “ED visits” listed twice. For both Kulldorff’s and Tango’s statistics, write “spatial scan statistic” the same way. I would suggest specifying the type of regression (e.g., linear regression).

Introduction:
i. Line 69: It should read, “a significant economic burden”.
ii. Line 76: Define ED here which is the first time it is used in the body of the manuscript.
iii. Line 78: It should read, “45 or older involved people with diabetes”.
iv. Lines 112-122: Remember to replace “risks” with “rates”.

Materials and methods:
i. Please see previous comments concerning “risks” vs. “rates” throughout the text.
ii. Line 136: It should read, “Florida’s population”.
iii. Lines 138-141: This might read better as follows, “By racio-ethno-cultural grouping, non-Hispanic White, non-Hispanic Black, Hispanic, and non-Hispanic other races make up {list values in order} of the population of Florida, respectively.”
iv. Line 183: It should read, “5-year average”.
v. Lines 196-199: In the results, the authors reported these statistics for every variable (i.e., Table 1) but indicated with an asterisk which variables were not normally distributed. It might be better to write, “means, standard deviations, median (50th percentile), and lower-upper quartiles were reported for all variables, but the authors have indicated where variables are non-normal and statistics based on quantiles should be used.”
vi. Lines 207 & 209: I am a bit confused if alpha=0.20 refers to the significance level/type I error risk or another parameter since they later state using a critical p-value of 0.05. Most statisticians prefer that alpha/significance level be reported in the methods (e.g., alpha=0.05 or a significance level of 5%) and then report the p-values in the results.
vii. Lines 211-212: This sentence needs some elaboration. A high-rate area would typically be an area whose rate ratio was greater than 1. It was unclear if the selection of 1.2 for defining a high rate was based on “epidemiological significance”. It was unclear if “very-low risk clusters” meant a rate ratio >1 but less than 1.2 or a rate ratio < 1. Please clarify in the revised document.
viii. Line 213: It should read, “Kulldorff’s”.
ix. Lines 203-221: Please note the rule used for reporting secondary clusters.
x. Line 230: Please include absolute value symbols around 0.7.
xi. Lines 238-240: Did the authors use a causal diagram to differentiate between confounding and intervening variables or only consider a 20% change in model coefficients?
xii. Line 241: I believe the text should read, “statistically significant ones were kept in the final model”.
xiii. Line 246: Please note if the inverse distance spatial weight was based on the centroids of the counties or another location.

Results:
i. Line 278: It should read, “per 100,000 person-years”.
ii. Line 310: A brief statement on the variables that were statistically significant on univariable analysis might be worth mentioning before describing the results in the final model. The authors may want to consider if the statistically significant variables on univariable analysis that were not included in their final model were the result of controlling for a confounder or the inclusion of an intervening variable in the model.
iii. Lines 310-317: I would recommend after indicating that the variable is statically significant and the direction of the association just referencing the table in parentheses to avoid repeating specific statistical results in the text and tables.
iv. Line 318: It might be worth commenting on diagnostics concerning individual observations (e.g., were there any concerns about outliers or highly influential observation)?
v. Line 329: It might read better as “counties had a high percentage of uninsured people.”

Discussion:
i. Line 358: It should read, “resulted”.
ii. Line 367: Please include a reference at the end of this sentence.
iii. Line 371: It might be better to write, “Black or Hispanic people”.
iv. Line 376: It should read, “and the percentage of married people”.
v. Line 379: It might be clearer to write, “This relationship is explained by married people being more…”
vi. Line 401: I would suggest writing, “Coding errors, leading potentially to misclassification bias, occur when using ICD-10…”
vii. Line 404: It might be clearer to write, “that used previously collected administrative data…”
viii. Line 410: It should read, “This study identified geographic…”
ix. Line 412-413: It is unclear where you discussed “low socio-economic status”. Perhaps it would be better to remove this phrase and replace it with “low levels of insurance coverage”.

References:
i. Please make certain the titles of journal articles are consistently formatted in terms of the use of upper and lower case letters.
ii. Line 443: It should read, “of diabetes”.

Tables & Figures:
i. In the Table 1 footnote, it should read “Standard deviation”.
ii. In the titles of Tables 2 & 3 and Figures 4 & 5, it might be best to write, “Statistically significant non-overlapping spatial clusters…”, and make certain to replace “risks” with “rates” in all tables and figures. If the locations of clusters are not going to be included in the tables, it might be good to include a footnote in the tables to the relevant figure.
iii. It is not clear why different headings are chosen for Tables 4 & 5. It would be adequate to have a column for the coefficient, 95% CI, and p-value for both tables.
iv. Figure 1. It should read, “square mile” in the legend.
v. Figure 3. It should read, “ED visits/100,00 person-years”.
vi. Figure 6. It might be clearer to write, “Distribution of statistically significant” and “Percent without health insurance”.

Reviewer 2 ·

Basic reporting

No comment.

Experimental design

The research questions are overall well defined, relevant and meaningful. However:
1. Line 136-147: I would expect to see them in the Results section instead of Methods?
2. It is mentioned in Line 163 that the BRFSS data only contains individuals aged 18 years or older, but it is not clearly stated if the study only included adult individuals.
3. Although stated in table 4 and table 5, it is not clearly stated in the manuscript that the models to identify predictors for diabetes-related ED visit risks were for year 2019. The reason of choosing year 2019 and discarding 2016-2018 data should also be clearly stated.
4. Was any multiple comparison adjustment done for table 4?

Validity of the findings

No comment.

Additional comments

No comment.

---

## Round 0.2 · Minor Revisions

Please revise your manuscript according to the reviewer's comments.
Yours,
Yoshi
Prof. Yoshinori Marunaka, M.D., Ph.D.

Reviewer 1 ·

Basic reporting

Overall, the manuscript meets these requirements and specific edits have been noted in my full review (see Additional Comments).

Experimental design

Overall, the manuscript meets these requirements and specific edits have been noted in my full review (see Additional Comments).

Validity of the findings

Overall, the manuscript meets these requirements and specific edits have been noted in my full review (see Additional Comments).

Additional comments

Manuscript ID: PeerJ-98661.R1

Title: Investigation of geographic disparities and temporal changes of non-gestational diabetes-related emergency department visits in Florida: a retrospective ecological study

Authors: Khan-MM & Odoi-A

General Comments:
Overall, this remains a strong manuscript of public health importance. In their responses and revisions, the authors have been able to meet most of my requests and answer my questions concerning the previous draft of their manuscript in a convincing manner. Unfortunately, in a few instances their arguments concerning certain requests/questions were unconvincing or the references provided to justify their response was inadequate or contradicted their argument.

1) Issue of risks vs rates:
I had assumed in the first draft the authors were using the terms risk and rate interchangeably. I had requested they change it to rate, but the authors indicated they felt these were truly risks and provided references, including one from Methods in Epidemiologic Research. However, those references supported my request to report the results as rates. Furthermore, one of the authors has published a similar manuscript on hospitalizations and non-gestational diabetes that uses similar data and they reported their findings as rates.

I suggest the authors review the following and make the appropriate corrections to the manuscript concerning risks and rates that were previously requested:

i. The section on rates of hospitalizations from the CDC’s National Diabetes Statistics Report (https://www.cdc.gov/diabetes/php/data-research/index.html).

ii. A recently published manuscript by one of the co-authors:
Lord J, Odoi A. Investigation of geographic disparities of diabetes-related hospitalizations in Florida using flexible spatial scan statistics: An ecological study. PLoS One. 2024 Jun 4;19(6):e0298182. doi: 10.1371/journal.pone.0298182.

iii. Chapter 4, Pages 79-80 of Methods in Epidemiologic Research (https://projects.upei.ca/mer/) cited in the response letter.

The following example taken from MER should clarify their confusion:
“If there are 30 cases of diarrhea in a 100-patient nursing home over a 3-month period, the incidence rate is 30/(100*3)=0.1 cases per person-month.”

They should note that in their manuscript, they are calculating the number of ED visits in a population within a one-year period. A risk is dimensionless, can only be between 0 and 1, and is often expressed as a percentage. In theory, a rate could exceed 1 case/person-year as demonstrated in the example on page 80 of MER.

2) Settings for scan statistics:
The authors provided references to support some decisions for settings used for their scan statistics that appeared somewhat arbitrary in the first draft of their manuscript. When I reviewed the references provided, these manuscripts did not always support their choices.

i. For instance, the authors state in their reply letter that “the recommendation while using Tango’s
flexible spatial scan statistics is to use a default searching window size of 15 counties.” This is
not actually stated in this manuscript (Tango T. A spatial scan statistic with a restricted likelihood
ratio. Japanese J Biometrics. 2008;29:75–95). However, in the 2005 manuscript by Tango and
Takahashi (Tango T, Takahashi K. A flexibly shaped spatial scan statistic for detecting clusters. Int J
Health Geogr. 2005 May 18;4:11), they do use 15 regions as the maximum size but there
justification is more related to a priori considerations. For instance Tango and Takahashi (2005)
make the following statements:

“In this example, we chose two kinds of maximum length K = 15 and K = 20 since it is not unreasonable to assume that an actual cluster size will be less than one third or one fourth of the size of the whole study area.”

“The current practical upperbound is around K = 30 for the reason that the execution time of our current algorithm will take more than a week if K > 30 for the number of regions m = 200 ~ 300. However, it seems to be unlikely that the length of the true cluster would be larger than 10 ~ 15 percent of the total number of regions. So, we think that our current algorithm can be applied to many epidemiological studies with small to moderate cluster sizes. However, for larger cluster sizes, a more sophisticated algorithm to increase the upperbound for K is needed.”
I would strongly recommend that the authors justify the size of the maximum cluster based on a priori considerations concerning the proportion of the region that could reasonably be part of the cluster as done by Tango & Takahashi (2005).

Similarly, in response to my comment that “the maximum window was set at 13.5% of the
population based on the population of the largest county in Florida to ensure that all counties have
a chance of being a cluster. However, scan statistics can identify smaller clusters if they are
statistically significant and using this setting prevented Miami-Dade County from being part of a
larger cluster”, the authors responded that “we agree with the reviewer that increasing the window
size might detect a large cluster including Miami-Dade. However, that cluster could potentially
include other regions with non-elevated risks. Another reason for using a small window size in
Kulldorff’s scan statistics is that Florida has irregular coastal lines on both the east and west.
Previous studies suggest that in areas with irregular shapes, a smaller window size should be used
in Kulldorff’s scan statistics to identify true clusters (Ribeiro SHR, Costa MA. Optimal selection of
the spatial scan parameters for cluster detection: A simulation study. Spat Spatiotemporal
Epidemiol. 2012;3: 107–120).” In fact, Ribeiro and Costa (2012) state the following:

“In general, there is not a consensus in the literature about a unique performance measure. Therefore, the most appropriate performance measure is related to the goal of the cluster analysis, that is, before running the cluster analysis the user might be conscious about what he or she wants to find. Alternatively, it can be argued that the results provided in this work can be used to draw guidelines for practical use of scan statistics. That is, given a study region, the user may concern first about the power statistic. In this case, the circular scan statistic or the elliptical scan statistic can be applied with a maximum cluster size of 50%. If the detected cluster is significant and small then the cluster analysis is complete. Otherwise, if the detected cluster is significant and large then the user may be interested in detecting a small group of areas more likely to belong to the true cluster. To do so, the user may re-run the scan statistic or the elliptical scan statistic with a maximum cluster size of 5%, or run the double scan statistic.”

I do not expect the researchers to use Ribeiro and Costa’s (2012) iterative approach, but the 13.5% is not well justified by the paper cited and within SaTScan there are “drilldown” options for examining the homogeneity of the rates/risks within a cluster. It might make more sense to use the <=50% default and discuss the uniformity of “risk” in the cluster, but it would also be fine to justify the choice based on a cluster size that would be plausible based on a priori knowledge and/or the descriptive statistics from the choropleth maps. I would avoid the current justification since it is not really supported by the cited manuscript.

3) Linear regression vs Poisson/negative binomial regression for risk factor analysis:
Based on their response letter, the authors appear to have modeled their data using both linear regression and negative binomial regression, achieved similar results, and elected to only report the analyses involving linear regression. While none of the model assumptions for their linear regression model were violated, their negative binomial model would have been preferable for modeling rates and would have accounted for the size of the populations within each county when estimating standard errors for model coefficients. If possible, I would recommend presenting the results of the negative binomial models and exponentiating the model coefficients to provide incident rate ratios, a measure of association that would be familiar to public health workers and epidemiologists.

Specific comments:
i. Please include the requested changes described in the general comments throughout the manuscript. The specific comments below are additional typographical or minor requests.

Abstract:
i. No additional edits required.

Introduction:
i. Line 102: It should read, “visits and hospitalizations.”
ii. Line 107: It should read, “a 54% increase.”

Materials and methods:
i. Line 135: Replace “Twenty two” with “Twenty-two”.
ii. Lines 136, 139 (2X): Replace “the rests” with “the rest”.
iii. Line 182: It should read, “A cartographic…”
iv. Line 186: It should read, “descriptive analyses”.
v. Line 210: It should read, “The cluster…”
vi. Line 212-213: Calling a cluster with an O/E, IRR, or RR greater than 1 a very low risk cluster does not really make sense. If the authors had run these cluster analyses as two-tailed tests, a low-risk cluster would have been a statistically significant cluster with an O/E, IRR, or RR less than 1. I would suggest that the authors either report all statistically significant non-overlapping clusters or report that “Only statistically significant non-overlapping clusters that had what we considered an epidemiologically notable higher rate of ED visits compared to the overall rate (i.e., O/E>1.2) were reported.”
vii. Lines 224: It should read, “Like our analyses using Tango’s method, the cluster….”
viii. Line 241-242: The authors should be a bit more specific concerning the phrase “biological and statistical considerations”. For instance, they could indicate that when dealing with highly correlated independent variables, they selected the variable with the lowest p-value and/or proximal relationship to the outcome.
ix. Line 248-250: The authors should clarify if they differentiated between confounding and intervening variables using a causal diagram.
x. Lines 250-252: The authors should be a bit more specific about what types of interactions would have been considered “biologically meaningful”.
xi. Line 253. It should read, “the variance inflation factor (VIF).”
xii. Line 253. In the results (see line 337), the authors indicate they used a cut-off for VIF of 10, but in the methods, they state they intended to use a cut-off of 5 for VIF. They should be consistent in the cut-off used throughout the document and VIF >= 10 is frequently used in the published literature as an indicator of collinearity.
xiii. Line 254. It should read, “The assumptions of homoskedasticity, normality, and spatial independence were assessed……”
xiv. The authors did not address the assumption of linearity. It might be adequate to say that “Non-linear relationships between county characteristics and the rate of EM visits for non-gestational DM were not considered in our analyses.”

Results:
i. Line 306: It might be better to write, “clusters were similar in size and location across years.”
ii. Line 307: Start the sentence with “Using Tango’s FSSS, the primary….” This addition will clarify which set of results are being described.
iii. Line 333: Add “and the following independent variables:” before the list of variables.

Discussion:
i. Line 392: It should read, “This difference was the result of non-Hispanic Black patients having……”
ii. Line 396: It should read, “minority groups, such as Black and Hispanic populations, are…..”
iii. Line 410: It should read, “reports from previous studies.”
iv. Line 411: Remove one of the periods at the end of the sentence.
v. Line 413: It should read, “kidney disease”.
vi. Line 427-428: It should read, “Coding errors might have occurred……”
vii. Line 429: It should read, “due to the complex……coding system that might….”
viii. Line 436-439: The authors have largely interpreted their results in the discussion at the individual-level so I would suggest they rewrite this sentence as follows: “It is important to note that since this study was conducted at the county-level, it is potentially subject to ecological bias so our inferences to the individual-level should be considered with caution.”
ix. Line 444. Include a period at the end of the sentence.

Tables and Figures:
Table 2. The column title RR may be the ratio of observed over expected (O/E). Please correct if necessary.
Tables 2 & 3. Please include a footnote for each table concerning settings used (e.g., maximum cluster size, one-tailed test, etc.).
Table 4. The title should read, “Results of univariable negative binomial regression models examining the associations between county characteristics and the rate of diabetes-related emergency department visits in Florida, 2019.”
Table 5. The title should read, “Results of a multivariable negative binomial regression model examining the associations between county characteristics and the rate of diabetes-related emergency department visits in Florida, 2019.”
Figure 6. I believe the data presented in the figure are just for 2019. If so, the authors should replace “2016-2019” with “2019” in the title.

Reviewer 2 ·

Basic reporting

no comment

Experimental design

Thank you for addressing the comments and the edits look good.

Validity of the findings

no comment

Additional comments

no comment

---

## Round 0.3 · accepted · Accept

Congratulations on the Acceptance.

Yours,
Yoshi
Prof. Yoshinori Marunaka, M.D., Ph.D.